# Microbial Diversity and Volatile Flavor Changes during *Gayangju* Fermentation, a Traditional Korean House Rice Wine

**DOI:** 10.3390/foods11172604

**Published:** 2022-08-27

**Authors:** Young-Ran Song, Byeong-Uk Lim, Sang-Ho Baik

**Affiliations:** Department of Food Science and Human Nutrition, Jeonbuk National University, Jeonju 561-756, Korea

**Keywords:** microbial diversity, volatile flavor, Korean traditional house rice wine, *Gayangju*

## Abstract

Physicochemical changes in fermented alcoholic beverages are significantly related to microbial community development during fermentation. Due to its unusually long fermentation, *Gayangju*, a traditional Korean house rice wine fermented with *nuruk* as the traditional starter, gives rise to a strong yeast community and, therefore, a high ethanol concentration and different flavors. However, no detailed analysis has been examined. Changes in microbial community structure during *Gayangju* fermentation were examined using both culture-dependent and culture-independent methods. During fermentation, *Saccharomyces cerevisiae* and *Saccharomycopsis fibuligera* were dominant during all stages of the fermentation. In contrast, Candida *parapsilosis*, *Hanseniaspora guilliermondii*, *Pichia anomala*, *Malassezia cuniculi* and *P. fermentans* were identified as minor. *P. anomala* appeared after the second brewing and then remained constant. Among the 19 compounds identified in this study as order-active compounds, 2-methyl-1-butanol (isoamyl alcohol) was the major compound that increased during the long fermentation stage. Most of the odor-active compounds such as 2,3-butanediol, 3-methyl-1-butanol, ethyl tetradecanoate, ethyl decanoate, ethyl dodecanoate, butanoic acid, 3-methylbutanoic acid (isovaleric acid), 2-methylbutanoic acid, 2-methyl-1-propanol, ethyl acetate, ethyl caprylate, 2-phenylethanol, and 3-methylbutyl acetate increased as the fermentation progressed during 68 days of fermentation, which showed significant differences in the concentrations of odor-active compounds of commercially fermented *makgeolli*.

## 1. Introduction

*Gayangju* is a Korean rice wine (*makgeolli*) traditionally home brewed since ancient times, and it has been widely consumed in Korea because of its deep taste and desirable flavor [1]. Compared to the typical commercial procedure for brewing Korean rice wine, which involves very short fermentation steps without *nuruk* as the fermentation starter, the *Gayangju* fermentation procedure is different regarding its unusually long fermentation steps, as shown in Figure 1 [2]. Due to its unusually long fermentation steps and aging processes—lasting two months with the addition of *nuruk* compared to commercial fermented rice wine, which takes only approximately five days without *nuruk*—*Gayangju* generally exhibits extraordinarily high ethanol concentrations of up to 18–20% (*w*/*w*) and chemical diversity including alcohols, esters, organic acids, fatty acids, and amino acids, which might lead to distinguished flavors and quality.

*Gayangju* is generally produced through three main processes: (1) alcohol fermentation with a natural starter culture (*nuruk*) at 25 °C for three days; (2) alcohol fermentation with the addition of newly prepared raw materials (mainly cooked rice) for two additional days; (3) post-process with aging without sterilization. Compared to commercial processes, this fermentation process largely affects not only the ethanol content but also the overall quality, including flavors. *Nuruk* is prepared by the natural solid fermentation of moistened ground rice or wheat sources, allowing for the complex mixed growth of molds (*Aspergillus*, *Rhizopus*, and *Mucor* spp.), yeasts (*Saccharomyces*, *Pichia*, *Candida*, *Torulopsis*, and *Hansenula* spp.), and lactic acid bacteria (*Leuconostoc*, *Pediococcus*, and *Lactobacillus* spp.) [3,4,5]. All these microorganisms can affect the physiological and biochemical properties of the final fermented products through metabolic pathways of alcohol and lactic acid fermentation after saccharification by molds that release amylase and glucoamylase [5]. Finally, the post-processing aging stage promotes the esterification between acids and alcohols, improving the flavor profile. Various volatile components developed via the alcoholic and lactic acid fermentation stage can be converted into small molecules, and yeasts can be used as basic materials for complex, volatile profiles affecting the physicochemical quality of *makgeolli* [5,6]. During the long period of *Gayangju* fermentation, in particular, unique microbial consortiums might result in different microorganism developments. Although many studies have shown microbial diversities in *makgeolli* or *nuruk*, there has been no information on the microbial diversity or biochemical properties of *Gayangju* during its fermentation process.

A few volatile compounds in commercially available *makgeolli* have been identified, consisting of 45 major volatile compounds composed of 33 esters, 8 alcohols, 1 aldehyde, 1 acid, 1 phenol, and 1 terpene. sp. [6,7]. The key aroma-active components are the esters such as ethyl decanoate, ethyl (Z)-octadec-9-enoate, ethyl octanoate, 2-phenethyl acetate, ethyl acetate, 3-methylbut-1-yl ethanoate, ethyl hexadecanoate, ethyl 9,12-octadecadienoate, ethyl dodecanoate, and ethyl tetradecanoate, usually known as by-products of yeasts [6,7]. The development of volatile compounds in *makgeolli* is influenced by several factors related to the fermentation process, including starting materials, processing techniques, and specific microorganisms. However, information regarding the volatile flavor compounds present in *Gayangju* and its change during fermentation is still lacking.

Thus, this study aimed to determine the microbial community succession and flavor change profiles during Korean traditional house rice wine *Gayangju* fermentation by adopting both culture-dependent and culture-independent methods. For the microbial community change analysis, a denaturing gradient gel electrophoresis (DGGE) method was used, and a gas chromatography-mass spectrometry (GC-MS) instrument with an automatic purge and trap concentrator was used for one-step direct flavor change analysis with low detection limits [8].

## 2. Materials and Methods

### 2.1. Gayangju Fermentation

*Gayangju* fermentation was carried out at the JeonJu Korean Traditional Wine Museum (http://urisul.net/ accessed on 6 June 2021). Rice (non-glutinous and glutinous rice) was obtained from Buan, Korea, and *nuruk* was acquired from Songhak Gokja (Gwangju, Korea), the representative manufacturing company for fermented *nuruk* production in Korea. Non-glutinous rice (1 kg) was washed and soaked briefly in distilled water at room temperature overnight and then ground. The rice cake made with boiled water was mixed with water (7 L) and *nuruk* (1.0 kg, 10% *w*/*w* of the total grain source). The mixture was placed in a crock and initially fermented at 25 °C for approximately 36 h until the temperature of the fermenting material reached 37 °C. After cooling to 10 °C for 36 h, the mixture was used as the starter material (first-stage mash), while glutinous rice (*Oryza sativa var. glutinosa* 8 kg) was washed, soaked overnight, and then steamed. The first-stage mash was further mixed with the heated glutinous rice, and the mixture was used as the second-stage mash for alcoholic fermentation. The second fermentation was conducted at 25 °C for 48 h. The final fermentation and aging stage was carried out at 15 °C for 35 days. Finally, the obtained *Gayangju* was filtered through filter cloth and bottled as the final product (Figure 1). The samples were collected every 6 h during the first fermentation, every 12 h during the second fermentation, and every week during the rest of the nine weeks (until the end of the 5-week fermentation and 4-week storing periods) to be analyzed during the index period. Sampling was done in triplicate from different jar fermentors. As a control, commercial rice wine was prepared as described below. A total of 200 g of nonglutinous rice was rinsed and then soaked in tap water for 3 h. After draining, the rice was immediately steamed for 40 min and quickly cooled by being spread out thinly on an aluminum pan. Commercial rice wine fermentation was carried out in a 1.5 L glass bottle along with the distilled water (300 mL), glucoamylase (800 GAU), α-amylase (1350 BAU), and *Saccharomyce cerevisiae* (equivalent to 10% of the total volume) at 25 °C

### 2.2. Physicochemical Analysis

During the fermentation, measurements of pH, total acidity, total soluble solids, residual sugar, and ethanol were performed. pH was measured using a pH meter (Orion model 710; Thermo, Beverly, MA, USA), and total acidity was measured with a 0.1 N NaOH solution. Total soluble solids were measured with a refractometer (PAL-α, 0–85 Brix; Atago, Tokyo, Japan), and the residual sugar content was analyzed using the DNS method (Miller, 1959). For the analysis of alcohol content, a GC system (HP 6890 series; Agilent Technologies, Waldbronn, Germany) and a J&W DB-5 capillary column (30 m × 0.25 mm id, 0.25 µm film thickness; J&W Scientific, Folsom, CA, USA) were used, with He as the carrier gas [9]. Isopropanol was used as the internal standard for the quantitation.

### 2.3. Free Amino Acids Analysis

For the free amino acid analysis, ethanol extraction was conducted with 70% (*v*/*v*) ethanol using a Branson 2510 ultrasonic bath (Branson Ultrasonics, Danbury, CT, USA) at 80 °C for 15 min. After eliminating the lipid phase by adding 20 mL of ether, the obtained solution was dried by evaporating under reduced pressure at 40 °C using a rotary evaporator (EYELA; Tokyo Rikakikai Co., Tokyo, Japan). The sample re-dissolved in sodium citrate buffer (pH 2.2) was analyzed with an amino acid analyzer with an LCAK60/Na cation separation column (150 × 4.6 mm). The column temperature was increased from 50 to 80 °C. Sodium citrate buffers (pH 3.3, 4.3, 5.2, and 10.1) and ninhydrin solution were used as mobile phases at flow rates of 50 and 25 mL/h, respectively.

### 2.4. Microbiological Analysis

For the counts of yeast, total aerobic mesophilic bacteria, and lactic acid bacteria (LAB), 100 μL of a decimal dilution in 0.85% sterile saline solution was spread onto YM agar plates supplemented with penicillin (20 units/mL), streptomycin (40 µg/mL, Sigma-Aldrich, St. Louis, MO, USA), nutrient agar (Merck, Darmstadt, Germany), and MRS agar (Difco, Franklin Lakes, NJ, USA) plates with cycloheximide (Sigma), respectively. Incubation was conducted at 29 °C for 48 h for yeast and at 37 °C for 30 h for total aerobic mesophilic bacteria and LAB. The number of viable cells was determined by counting in triplicate, and the results were expressed as log cfu/mL. For the PCR-DGGE analysis as a culture-independent method, DNA was extracted using a NucleoSpin^®^ Food genomic DNA extraction kit (Macherey-Nagel, Duren, Germany), following the manufacturer’s protocol for the purification of total genomic DNA. For the analysis of fungal diversity, fragments of the fungal gene at the 26S rRNA D1/D2 region were generated using the eukaryotic universal primer NL1 containing a CG-clamp and LS2 [9]. For the analysis of bacterial diversity, the V3 region of 16S rDNA was amplified by PCR using the universal bacterial primer 357F containing a CG clamp and 517R. DGGE analysis was performed in 8% (*w*/*v*) polyacrylamide gels (acrylamide: bisacrylamide 37.5:1) using a Dcode apparatus (BioRad, Richmond, CA, USA). A denaturing gradient gel from 30% to 60% was run at 125 V and 60 °C for 5.5 h and stained with an EtBr solution (0.5 µg/mL). Bands visualized under UV light (ChemiDoc XRS Imaging System, Bio-Rad) were re-amplified using the same primers without the GC clamp. The PCR products were purified using a QIAquick PCR purification kit (Qiagen, Hilden, Germany) and then analyzed with an automated DNA sequencer (ABI PRISM 3700; Applied Biosystems, Foster City, CA, USA). The identity of the sequences was determined by the BLASTN algorithm in the GenBank database.

### 2.5. Volatile Flavor Analysis

Volatile flavor compounds were analyzed using a GC/MS QP 2010 plus (Shimadzu, Kyoto, Japan) with an automated Purge & Trap Sampler JTD-505III (Japan Analytical Industry, Tokyo, Japan). In addition, GC-MS analysis for volatile flavor compounds was performed as described by Song et al., with 2-Methyl-3-heptanone used as an internal standard [10].

### 2.6. Statistical Analysis

All analyses were performed in triplicate. The data were analyzed using the SPSS ver. 16.0 program (SPSS Inc., Chicago, IL, USA). Statistical evaluation was performed using the one-way ANOVA, and significant differences were determined using Duncan’s multiple range tests at *p* < 0.05. The correlation between variables was determined by Pearson’s correlation analysis.

## 3. Results and Discussion

### 3.1. Physicochemical Change during Gayangju Fermentation

During the traditional Korean *makgeolli* process by the second brewing (fermentation for 42 days and storage for 14 days), the total acidity, pH, and alcohol and sugar contents were investigated. Korean commercial rice wines exhibited an acidic pH of 3.5–4.5 and a total acidity of 0.35–0.70 [8]. As the key factors related to fermentation properties, the overall pH and acidity could be changed by the production of various organic acids by the microbes—mainly, LAB strains. In this study, the total acidity rapidly increased to 0.69% during the first brewing process of three days, and then, after adding a second rice source, the acidity (0.24%) increased to 0.55% during the second brewing and aging periods (Figure 2A). Likewise, the initial pH of 6.52 sharply decreased to 3.68, and the pH value remained at 4.07–4.35 for the rest of the fermentation process. During the first brewing process, glutinous rice was saccharified, especially on the first day, and the reducing sugar content increased from the initial 3.7% to 4.2% on day 1 and then decreased to 2.6%, while the alcohol content rapidly increased to 17.4% (Figure 2B). Thereafter, the sugar content was again increased to 5.5% by adding a second rice source; however, this decreased to 4.3% on day 6. In contrast, the alcohol content (6.10%) on day 3 again increased to 16.0% during the second brewing process, and then the aging periods yielded a further increase to 19.2% on day 40. In addition, refrigeration storage for 12 days resulted in a further increase in acidity to 0.73%, and on day 70, the alcohol content decreased to 18.7%. Additionally, the sugar content rose slightly to 5.2%. In this study, the composition and contents of the free amino acids in the samples were also analyzed, and the results are shown in Figure 3. A significant difference in the total free amino acid contents between *Gayangju* (26.4 mg/mL) and commercial *makgeolli* (11.12 mg/mL) was observed. The contents of glutamic acid and aspartic acid (associated with flavors of ‘richness’), as well as threonine and alanine (associated with the taste of ‘sweetness’), were higher in *Gayangju* than they were in commercial *makgeolli*. In addition, the arginine and leucine (associated with flavors of ‘bitterness’) were significantly higher than commercial *makgeolli* (*p* < 0.01). Notably, *Gayangju* showed about two-times-greater contents of aspartic acid, which is the main influence on savory flavors compared to commercial *makgeolli*. Additionally, threonine, serine, and alanine, which are the main influences on sweetness, were 1.5- to 2-fold higher in the samples of *Gayangju*. The levels of tryptophan were significantly higher in *Gayangju*, which is known to occur only in fermentation and may be a potential precursor of an aroma compound, 2-aminoacetophenone (AAP). Although an increase in AAP is significantly related to the ‘untypical aging off-flavor’ (UTA) during fermentation, it seems that the high-level accumulation of tryptophan indicates no AAP progress [11].

### 3.2. Microbial Change during Makgeolli Fermentation

The populations of yeast, LAB, and total aerobic mesophilic bacteria during the fermentation and storage of *makgeolli* were estimated by the plating method (Figure 4). The LAB population numbers were similar to the total aerobic mesophilic bacteria during the overall fermentation and storage periods. Initially, the LAB count was 7.59 log cfu/mL; after 24 h of fermentation, the population reached 9.34 log cfu/mL and remained at similar levels until day 14. Through the continuous aging process, the LAB population began to decline, and a LAB count of 7.18 log cfu/mL was detected on day 42. However, the further storage periods yielded an increase to 7.53 log cfu/mL on day 70. At the beginning of the fermentation, the yeast population was approximately 7.70 log cfu/mL and increased over the following days, reaching the maximum population during the second brewing (9.09 log cfu/mL on day 4). After this time, the yeast population remained for two days and then decreased steadily until further aging and storage periods. On the 70th day, the yeast population was 6.16 log cfu/mL.

By a culture-independent method, the species composition and the dynamics of the fungal and bacterial community during the different fermentation and storage times were investigated (Figure 5), and a wide diversity of bacterial species existed in all the processes. Of the 17 bands analyzed, the DNA sequences of ten bands corresponded to LAB, revealing that it was the major bacterial group in *makgeolli* fermentation. Mainly, *Lactilactobacillus curvatus* (band 3), *Liquorilactobacillus satsumensis* (band 4), and *Pediococcus acidilactici* (band 5) were found to be predominant during the entire process, including the storage period. The species appeared as inferior bands in *nuruk* at day 0 but as dominant bands after one day. On the other hand, *Lactiplantibacillus plantarum* (band 1) was detected after one day and remained as a band with weak intensity over the following days. *P. pentosaceus* (band 8) was consistently detected as one of the marginal species throughout the overall fermentation process. In addition, the bands corresponding to *L. sakei* (band 2), *L. satsumensis* (band 15), *Lacticaseibacillus casei* (band 16), and *Lacticaseibacillus paracasei* (band 17) also persisted throughout the fermentation process. As other bacteria genera, *Erwinia* sp. (band 6), *Pantoea* sp. (bands 7, 9, 10, 14), *Enterobacter* sp. (bands 11, 13), and *P. inopinatus* (band 12) were identified in the *nuruk* sample but disappeared as the fermentation progressed.

Generally, the fungal communities during rice wine fermentation were far simpler than the bacterial communities (4). To date, several species of *Aspergillus*, *Rhizopus*, *Candida*, *Saccharomyces*, *Wickerhamomyces*, and *Saccharopmycopsis* sp. have been found in the spontaneous fermentation of Korean rice wines, of which *S. cerevisiae* was the dominant yeast throughout the entire process, representing an average of above 90% [5,12]. Meanwhile, the fungal community of the *nuruk* (the starter) was more complex, indicating the diverse mycofloral dynamics of *Aspergillus*, *Cladosporium*, *Eurotium*, *Lichtheimia*, *Mucor*, *Penicillium*, and *Rhizopus* sp., among others; however, they were distributed less than 0.1% [4,13]. Unlike rice wine, *Pichia* sp. was the most dominant yeast, whereas a representative alcohol fermentation strain, *S. cerevisiae,* was detected in only some of the *nuruk* samples [13]. In this study, fungal communities during the *makgeolli* fermentation were also assessed, and PCR-DGGE based on the analysis of 26S DNA clone libraries resulted in a total of 12 bands (Figure 5B). Yeast species including *Saccharomyces cerevisiae* (bands 1, 5, 10), *Candida parapsilosis* (band 2), *Hanseniaspora guilliermondii* (band 3), *Saccharomycopsis fibuligera* (bands 7, 8), *Pichia anomala* (band 4), *Malassezia cuniculi* (band 6), and *Pichia fermentans* (band 9) were identified, in which *S. cerevisiae* and *S. fibuligera* were found to be dominant, observed in all stages of the fermentation. Specifically, *S. cerevisiae* (band 1) also appeared as the dominant band in the *nuruk* material, and as other bands corresponded to *S. cerevisiae,* band 5 disappeared after entering the aging stage, while band 10 appeared during the first brewing process. In the case of *S. fibuligera*, band 8 was observed in the *nuruk* at all stages of the *makgeolli* process, while band 7 appeared during the first brewing process but remained as a superior band. In addition, *P. anomala* (band 4) appeared after the start of the second brewing and then remained constant; however, its band intensity was weak. With respect to the other strains, *C. parapsilosis* (band 2), *H. guilliermondii* (band 3), *M. cuniculi* (band 6), and *P. fermentans* (band 9) were detected as rare, minor strains. As mold sources, bands corresponding to *Rhizopus* sp. (band 11) and *Aspergillus oryzae* (band 12) were also identified in the *nuruk* material but disappeared during the first brewing. Most of the detected fungi in this research were reported in many kinds of fermented rice liquors. Meanwhile, no significant variation was observed in their community after the start of the storage. In rice wine fermentation, saccharification in the early stages is mainly due to molds from *nuruk* [5,12]. However, these molds are numerically inferior to yeast and constitute below 5% of the total microbe, and then the molds rapidly decrease during the fermentation process [5,12]. Meanwhile, *S. fibuligera* commonly exists in amylolytic yeast in Indigenous food fermentation using starchy substrates such as rice and cassava [14]. Although it is also thought of as a foodborne and dimorphous yeast, *S. fibuligera* has received increasing attention, as it can secrete amylase, glucoamylase, β-glucosidase activity, and trehalose [14]. Jung et al. (2012) reported that *S. fibuligera* existed at high levels during the increase in glucose concentration and was replaced by *S. cerevisiae* as the ethanol concentration increased. *S. fibuligera* is also used as the main amylase producer for ethanol production from starch [14]. *P. anomala* has a positive role in food preservation and is well known as a flavor-enhancing (especially ester-producing) yeast in food and beverage fermentation [12]. In recent years, the significance of non-*Saccharomyces* species in winemaking has attracted the interest of winemaking researchers, and they contribute to the final taste and flavor of wines. Some non-*Saccharomyces* yeasts, such as *Candida* sp., can negatively affect the aroma and flavor of wine [15].

### 3.3. Volatile Compounds Change in Gayangju

Flavor and aroma are essential distinguishing characteristics of fermented wine and result from raw materials, winemaking practices, yeast strains, and aging conditions [16]. Moreover, most of the volatile compounds responsible for the organoleptic characteristics in rice wine are directly or indirectly transformed from the constituent of rice metabolism [17]. The increment values of major order-active flavor components during *Gayangju* fermentation are shown in Table 1. Among the 19 compounds identified in this study as order-active compounds, 2-methyl-1-butanol (isoamyl alcohol) was the major compound that significantly increased during the long fermentation stage. Most of the odor-active compounds such as 2,3-butanediol, 3-methyl-1-butanol, ethyl tetradecanoate, ethyl decanoate, ethyl dodecanoate, butanoic acid, 3-methylbutanoic acid (isovaleric acid), 2-methylbutanoic acid, 2-methyl-1-propanol, ethyl acetate, ethyl caprylate, 2-phenylethanol, and 3-methylbutyl acetate increased as the fermentation progressed during the 68 days of fermentation, showing significant differences in the concentrations of odor-active compounds of commercially fermented *makgeolli*. PCA and HCA analyses were performed to understand the volatile flavor development change during the long fermentation stage of *Gayangju*. As shown in Figure 6, an overall PCA biplot was constructed with a total variance of 80.34%. PC1 was strongly and positively related to most volatile compounds occurring in the late stage of *Gayangju* fermentation, while it was strongly negatively related to the early stage of *Gayangju* fermentation. In addition, most of the volatile compounds are positively related to *Gayangju* and not commercially produced rice wine. Based on these properties, the *Gayangju* fermentation at different stages was differentiated with GY3, GY5, GY40, and GY68 positioned fully in the strongly positive PC1 region. The later aging stage samples were positioned in the positive PC1 region due to the elevated concentrations of most order-active compounds. During the spontaneous fermentation of wine, yeast produces higher alcohols and esters, and the compounds strongly influence the sensory properties of the resulting wine [16]. In this study, alcohols and esters also comprised the largest groups of compounds in rice wines. Most identified volatile compounds were common in many other fermented alcoholic beverages such as wine, beer, and rice wine [18,19]. Compounds including 2-methyl-1-propanol, 2-methyl-1-butanol, 3-methyl-1-butanol, 3-methylbutyraldehyde, ethyl acetate, and 3-methylbutyl acetate were present in all rice wine samples. 2-methyl-1-butanol (fermented and malt-like order notes) as amyl alcohols usually converted from leucine through deamination and decarboxylation reactions during the long-term aging fermentation of wine, beer, and sake [20,21]. The significantly increased concentrations of 2-methyl-1-butanol in the long-term aging period of *Gayangju* could be related to the qualities of alcoholic beverages due to their characteristic fermented, malt-like, and alcoholic-like odor notes [20]. The commercial rice wine prepared by *S. cerevisiae* strain (Lalvin EC-1118) as an ethanol producer resulted in less small, diverse compounds of eight, with the lowest concentration of total compounds. Mainly, 2-methyl-1-propanol (isobutyl alcohol; sweet; fusel, spirituous, sweet pear/nutty) was only detected as a major compound in commercial rice wine, in which fusel alcohols are formed from branched-chain amino acids by yeast during alcohol fermentation [17]. *S. cerevisiae* is an essential fermentative microorganism that produces ethanol from glucose, fructose, and sucrose. It synthesizes both nutritive (amino acids and vitamins) and flavor-volatile compounds, such as ethyl esters (ethyl decanoate, ethyl dodecanoate, and ethyl tetradecanoate) that influence the quality and aromatic profile of beverages [12].

## 4. Conclusions

This study was the first report to reveal the dynamics of microbial succession and the changes in flavor compounds during the long fermentation of Korean house rice wine, *Gayangju*. During fermentation, *Gayangju* maintains a strong yeast community, with a high ethanol concentration of 18.7%. *Saccharomyces cerevisiae* and *Saccharomycopsis fibuligera* were dominant at all stages of the fermentation, even in high ethanol circumstances. However, the non-saccharomyces of *Candida*
*parapsilosis*, *Hanseniaspora guilliermondii*, *Pichia anomala*, *Malassezia cuniculi*, and *P. fermentans* were minor. The major bacterial groups in the *Gayanju* fermentation were *Lactilactobacillus curvatus*, *Liquorilactobacillus satsumensis*, and *Pediococcus acidilactici* during the entire process, including the storage period. The volatile compounds analyzed by a GC/MS with an automated Purge & Trap Sampler were made of 19 compounds, including 2-methyl-1-butanol (isoamyl alcohol) as the major compound that increased during the long fermentation stage. The significantly increased concentrations of 2-methyl-1-butanol in the long-term aging period of *Gayangju* could be related to the qualities of *Gayangju* with malt-like and alcoholic-like odor notes. Further studies regarding the relationships between the flavor profiles and the specific microbes can bring us closer to improving the quality of *Gayangju* and the efficiency of its production.

## Figures and Tables

**Figure 1 foods-11-02604-f001:**
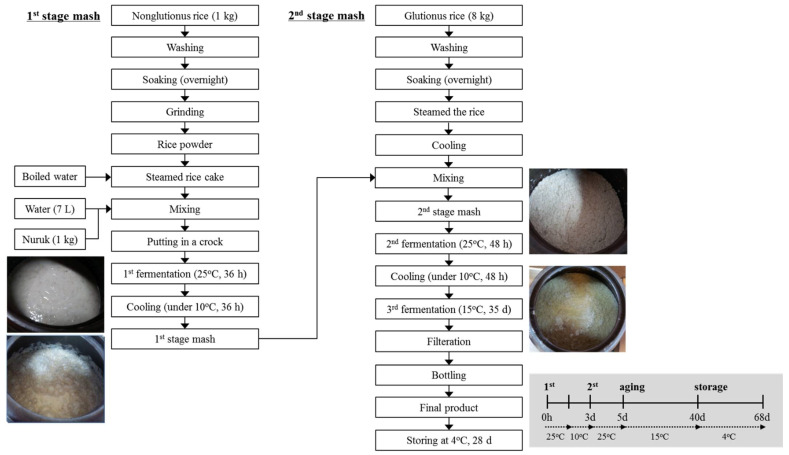
Flow sheet depicting the preparation of fermented traditional Korean rice wine, *Gayangju*.

**Figure 2 foods-11-02604-f002:**
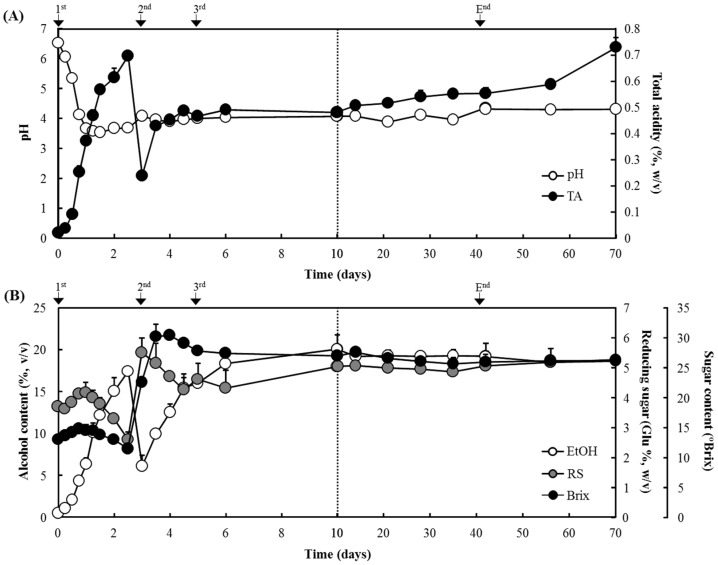
Changes in physicochemical characteristics during traditional Korean rice wine (*Gayangju*) fermentation. (**A**) pH (
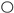
) and total acidity (●); (**B**) contents of alcohol (
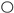
), reducing sugars (
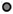
), and total soluble solids (●).

**Figure 3 foods-11-02604-f003:**
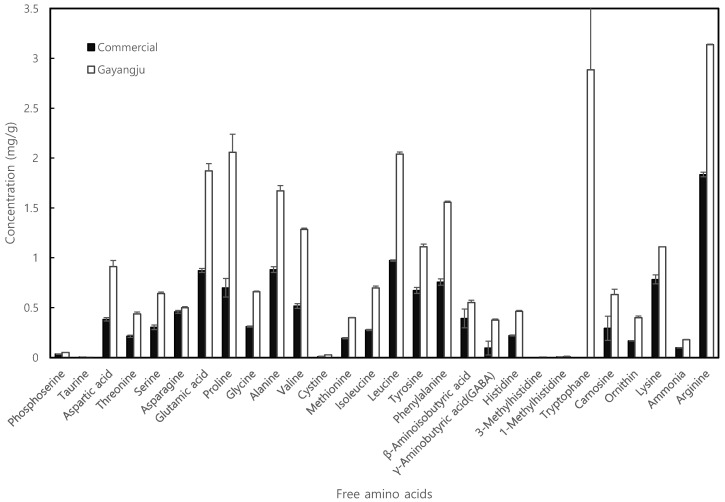
Free amino acids analysis of *Gayangju* and the commercial rice wine *makgeolli*.

**Figure 4 foods-11-02604-f004:**
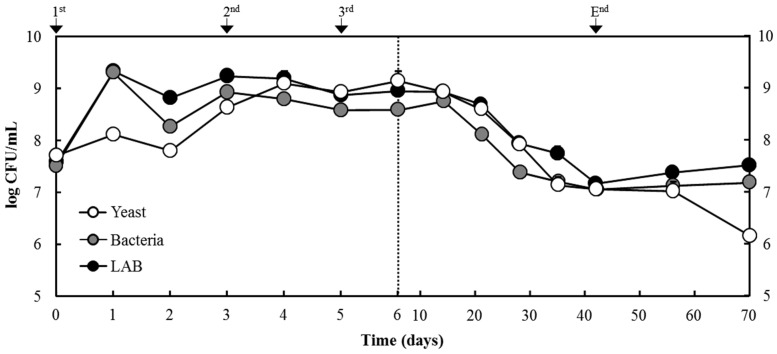
Changes in microbial counts (yeast, total bacteria, and lactic acid bacteria) during traditional Korean rice wine fermentation.

**Figure 5 foods-11-02604-f005:**
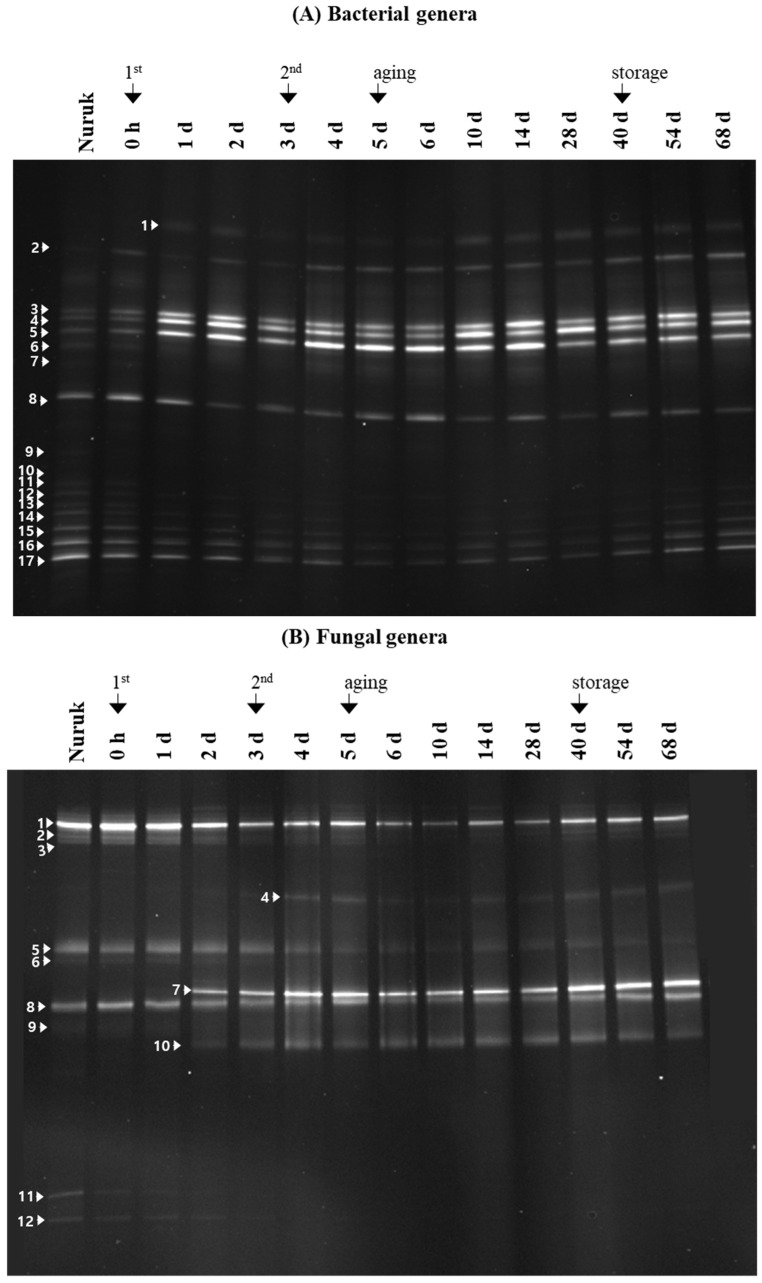
DGGE fingerprints of bacterial 16S rDNA (**A**) and eukaryokic 26S rDNA (**B**) during Korean traditional rice wine fermentation. The closest relatives of the fragments sequenced compared to the sequences retrieved from the GenBank database are as follows: (A1) *Lactiplantibacillus plantarum* (AY590777), (A2) *Lactilactobacillus sakei* (JN851763), (A3) *Lactilactobacillus curvatus* (JQ247525), (A4) *Liquorilactobacillus satsumensis* (AB362684), (A5) *Pediococcus acidilactici* (AB627837), (A6) *Erwinia* sp. (KC853200), (A7) *Pantoea* sp. (DQ122375), (A8) *Pediococcus pentosaceus* (AB236655), (A9) *Pantoea agglomerans* (DQ122373), (A10) *Pantoea* sp. (JF946788), (A11) *Enterobacter cowanii* (JQ660056), (A12) *Pediococcus inopinatus* (JN863658), (A13) *Enterobacter cloacae* (KF481919), (A14) *Pantoea* sp. (DQ122350), (A15) *Liquorilactobacillus satsumensis* (AB362684), (A16) *Lacticaseibacillus casei*, (A17) *Lacticaseibacillus paracasei*; (B1) *Saccharomyces cerevisiae*, (B2) *Candida parapsilosis*, (B3) *Hanseniaspora guilliermondii*, (B4) *Pichia anomala*, (B5) *Saccharomyces cerevisiae* (GU080046), (B6) *Malassezia cuniculi* (GU733708), (B7) *Saccharomycopsis fibuligera* (JX141337), (B8) *Saccharomycopsis fibuligera* (HM107786), (B9) *Pichia fermentans* (JQ665247), (B10) *Saccharomyces cerevisiae* (JX;141338), (B11) *Rhizopus* sp., (B12) *Aspergillus oryzae*. The similarity of all band sequences was ≥ 97% compared with those available in the GenBank database.

**Figure 6 foods-11-02604-f006:**
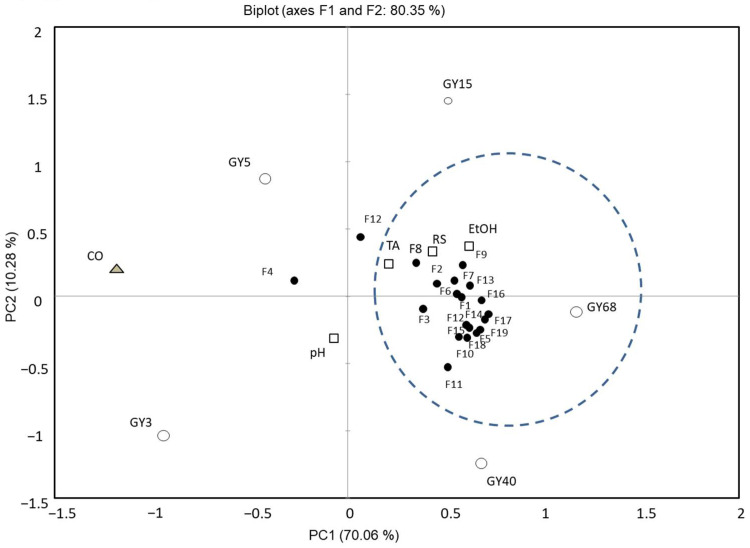
Biplot of the principal component analysis of volatile compounds from Korean traditional fermented *makgeolli*, *Gayangju*. pH, TA, RS, and EtOH indicate final pH, total acidity, soluble sugar, reducing sugar, and ethanol, respectively (□); The codes for flavor compounds (●) are defined in Table 1; As fermented *Gayangju* (
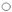
, GY), GY3, GY5, GY15, GY40, and GY68, indicate samples of different fermentation time of *Gayangju*. CO (
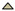
) indicates commercial *Makgeolli*.

**Table 1 foods-11-02604-t001:** Changes in volatile flavor compounds in *Gayangju* fermentation.

No.	Flavor Compounds	Relative Peak Area					
		Commercial *Makgeolli*	*Gayangju Makgeolli*				
			3 Days	5 Days	15 Days	40 Days	68 Days
F1	2,3-Butanediol	0	1 ^a^	2.15 ^b^	5.79 ^c^	5.43 ^c^	9.41 ^d^
F2	3-Methyl-1-butanol	2.97 ^a^	1 ^b^	2.09 ^c^	4.59 ^d^	3.56 ^e^	7.29 ^f^
F3	2-Methyl-1-butanol	0	1 ^a^	2 ^a^	6.79 ^b^	59.35 ^c^	691.53 ^d^
F4	2-Methyl-1-propanol	5.79	0	0	0	0	0
F5	Ethyl tetradecanoate	0	1 ^a^	1.59 ^b^	3.02 ^c^	4.53 ^d^	4.92 ^d^
F6	Ethyl dodecanoate	0	1 ^a^	2.20 ^b^	5.26 ^c^	5.15 ^c^	5.54 ^d^
F7	Ethyl decanoate	0	1 ^a^	2.36 ^b^	5.42 ^c^	4.45 ^d^	8.65 ^e^
F8	butanoic acid	0	1 ^a^	3 ^b^	16.2 ^c^	4.05 ^b^	9.11 ^d^
F9	2-Methylbutanoic acid	0	1 ^a^	2.74	5.71	4.74	8.13
F10	3-Methylbutanoic acid	0	1 ^a^	2.67	1.9	5.37	8.63
F11	2-Methylbutylaldehyde	0.85 ^a^	1 ^a^	1.2	1.25	3	2.75
F12	3-Methylbutylaldehyde	1.60 ^a^	1 ^b^	1.67 ^a^	1.87 ^a^	0.23 ^b^	2.04 ^a^
F13	2-Methylthio-1-propanol	0	1 ^a^	2.51 ^b^	5.45 ^c^	4.89 ^c^	6.59 ^d^
F14	Ethyl acetate	0.69 ^a^	1 ^a^	1.2 ^a^	1.7 ^b^	2 ^b^	2.2 ^b^
F15	Ethyl caprylate	0.1 ^b^	1 ^a^	1.96 ^c^	2.38 ^d^	4.29 ^e^	6.19 ^f^
F16	2-phenylethylethanlol	0	1 ^a^	2.23 ^b^	5.11 ^d^	4.32 ^c^	8.49 ^e^
F17	Ethyl pentadecanoate	0	1 ^a^	1.22 ^a^	2.81 ^b^	2.69 ^b^	3.12 ^c^
F18	3-Mehtylbutyl acetate	0.13 ^a^	1 ^a^	1.74 ^b^	2.74 ^c^	4.07 ^d^	8.15 ^e^
F19	2-Ethyl-1-hexene	0.07 ^a^	1 ^a^	0.8 ^a^	6.07 ^b^	8.11 ^c^	9.43 ^d^

The increment values are calculated by the number of volatile flavor compounds in other samples divided by the number of volatile flavor compounds in a 3-day fermented *Gayangju*. Different small letters (a–f) indicate significant differences of values between commercial *Makgeolli* samples and *Gayangju Makgeolli* with different fermentation time (*p* < 0.05).

## Data Availability

The data presented in this study are available in the manuscript.

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
