# Peer review of "Microbial Diversity and Volatile Flavor Changes during Gayangju Fermentation, a Traditional Korean House Rice Wine"

_foods, 2022, doi:10.3390/foods11172604_

Round 1
Reviewer 1 Report
In this article, the author described the composition of microbial and aroma profile a traditional Korean rice wine Gayangju. There are some major flaws as below:
1. The writing needs to be refined, as the description in article is repetitive and unclear.
2. Some concept was vague, for instance the reducing sugars in article was not defined, it is hard to know how it was calculated; the difference of Gayangju and commercial makgeolli was not clearly described; control group commercial was introduced but not explained.
3. The results of research are relatively weak related to how it was descripted in article. DGGE analysis of microbial composition was able to demonstrate the existence of microorganism, but not to decide if a yeast or bacteria is dominant or not during fermentation; the aroma by GC-MS was only demonstrated as peak area, without quantification and OAVs the aromas cannot be described as of importance.
4. Much description of results but not enough conclusion was provided. Figure 2. demonstrated the change of physiochemical parameters but fails to draw any conclusions, and the same issue happens to Figure 4. The change of aroma in different stage was almost neglected, whereas it could be interesting to discuss together with how microbial and physiochemical parameters change.
5. The conclusion part is poorly written, more like an unfinished discussion. Clear point on what results were obtained and what conclusions can be conducted should be made.
Author Response
To Reviewer 1
Q1. The writing needs to be refined, as the description in article is repetitive and unclear.
A1. Thanks for your valuable comments for our manuscript. We refined our manuscript as your indications
Q2. Some concept was vague, for instance the reducing sugars in article was not defined, it is hard to know how it was calculated; the difference of Gayangju and commercial makgeolli was not clearly described; control group commercial was introduced but not explained.
A2. As indicated in Materials and Methods in our manuscript (P4 L110), we measured residual sugars during fermentation to know fermentation activity by using DNS methods, which generally used broadly.
A2-1 Concerning on the difference of Gayangju and commercial makgeolli was described at Introduction at second sentence. To make clear, we added “commercial procedure for brewing Korean rice wine, which done very short fermentation steps without nuruk as the fermentation starter,” into the sentence. We also newly inserted preparation method for commercial procedure for korean rice wine used in this study at materials and methods (P3 L106-111).
Q3. The results of research are relatively weak related to how it was descripted in article. DGGE analysis of microbial composition was able to demonstrate the existence of microorganism, but not to decide if a yeast or bacteria is dominant or not during fermentation; the aroma by GC-MS was only demonstrated as peak area, without quantification and OAVs the aromas cannot be described as of importance.
A3-1. Our purpose in this study is to know how Gayangju fermentation are different in aspect microbial diversity and flavors compared to commercial rice wine which usually prepared by using single fungus of Aspergillus oryzae, saccharifying enzyme (amylase et al) and Saccharomyce cerevisiae. As mentioned in Results and Discussions, although we did not identify every colonies during every fermentation steps, we can understand microbial changes by both results of DGGE and microbial counts during fermentation of Gayangju. Moreover, DGGE analysis can be a generally established method to estimate dominate strains by comparing band strength.
A3-2 Most of the volatile compounds detected in this study already known compounds at other studies related with Korean rice wine (Ref 11 and 20). We selected main volatile compounds based on those references and compared.
Q4. Much description of results but not enough conclusion was provided. Figure 2. demonstrated the change of physiochemical parameters but fails to draw any conclusions, and the same issue happens to Figure 4. The change of aroma in different stage was almost neglected, whereas it could be interesting to discuss together with how microbial and physiochemical parameters change.
A4. From Figure 2and 4, we can understand about Gayangju fermentation progress during long fermentation compared to commercial fermentation process. Moreover, we could understand long fermentation of Gayangju clearly about long fermentation affects free amino acid compositions. We also showed how the aroma change occurred during Gayangju fermentation as shown in PCA analysis
Q5. The conclusion part is poorly written, more like an unfinished discussion. Clear point on what results were obtained and what conclusions can be conducted should be made.
A5. Thanks for your advice. We rewritten our conclusion with main results.
Reviewer 2 Report
Dear Authors,
I have carefully read the submitted paper. The choose of the subject – Microbial Diversity and Volatile Flavor Changes during Gayangju Fermentation, a Traditional Korean House Rice Wine - seems to be very justified, especially for the aim of exploring fermentation both from microbiological and chemical point of view. In the literature, there is very few well documented data concerning this specific traditional makgeolli, which gives interest to the study. However, crucial information is missing and must be added to make the article suitable for publication.
The major drawback of this study is the fragility of knowledge coming from one single fermentation, without biological replicates (only Analysis that were performed in technical triplicate). Authors should repeat the fermentation at least another 2 times, otherwise the obtained results are not describing Gayangju Fermentation, but only a specific batch of Gayangju happened to ferment in their lab. Data from 2 more experiments should be added and properly discussed (after statistical analysis which will be, at that point, possible) to make the paper suitable for publication.
Similarly, Authors do not describe, in the Materials and Methods section or elsewhere, which commercial makgeolli they compare with their samples. Assuming there are hundreds of commercial products, differing one from another to a certain extent, they should describe and reference the one chosen, and justify why it is representative of the whole beverage.
Another important point is that taxonomic annotation of bacteria after DGGE should be updated, as, for instance Lactobacillus plantarum does not exist anymore since the 2020 reclassification, in favor of Lactiplantibacillus plantarum, and so many other former-Lactobacilly A taxonomic note on the genus Lactobacillus: Description of 23 novel genera, emended description of the genus Lactobacillus Beijerinck 1901, and union of Lactobacillaceae and Leuconostocaceae | Microbiology Society (microbiologyresearch.org)
https://www.microbiologyresearch.org/content/journal/ijsem/10.1099/ijsem.0.004107
Some more specific remarks:
- Line 88: please add information about glutinous and non-glutinous rice (variety, commercial name) and clarify the meaning (vs gluten presence)
- Lines 125-130: incubation conditions do not include microaerophily or anaerobiosis, therefore LAB isolation is partial. Authors should discuss this point. Please also provide antibiotics concentrations.
- Figure 4: please discuss total bacterial counts being lower than LAB-only counts.
- Line 251: please add a reference
- Lines 276-277: please add proper references.
- Lines 294-300: it is not clear how this part of discussion brings value to the paper. Since the literature about non-Saccharomyces yeasts in wine is huge and much wider than the single 11-years-old reference cited, either the Authors provide a recent and insightful reading of the point and clearly link it with their results, or (suggested) remove this part.
- Line 334: no statistics appear in the paper. Please explain how significance is determined.
Author Response
To Reviewer 2
Q1 I have carefully read the submitted paper. The choose of the subject – Microbial Diversity and Volatile Flavor Changes during Gayangju Fermentation, a Traditional Korean House Rice Wine - seems to be very justified, especially for the aim of exploring fermentation both from microbiological and chemical point of view. In the literature, there is very few well documented data concerning this specific traditional makgeolli, which gives interest to the study. However, crucial information is missing and must be added to make the article suitable for publication.
A1 Thanks
Q2 The major drawback of this study is the fragility of knowledge coming from one single fermentation, without biological replicates (only Analysis that were performed in technical triplicate). Authors should repeat the fermentation at least another 2 times, otherwise the obtained results are not describing Gayangju Fermentation, but only a specific batch of Gayangju happened to ferment in their lab. Data from 2 more experiments should be added and properly discussed (after statistical analysis which will be, at that point, possible) to make the paper suitable for publication.
A2 We did three times not only for technical triplicate but did not mentioned in the manuscript. As mentioned in Material and Methods, fermentation did not carried out in our lab. Gayangju fermentation was done at JeonJu Korean Traditional Wine Museum (http://urisul.net/) where regularly fermented Gayangju and we sampled three times from three different jar fermentors. Thus we inserted “Sampling was done in triplicate from different jar fermentor.”
Q3 Similarly, Authors do not describe, in the Materials and Methods section or elsewhere, which commercial makgeolli they compare with their samples. Assuming there are hundreds of commercial products, differing one from another to a certain extent, they should describe and reference the one chosen, and justify why it is representative of the whole beverage.
A3 Yes, there are hundreds of commercial products. Thus we select most generally used methods as indicated in Materials and methods. To make clear we inserted preparation procedure for commercial rice wine.
Q4. Another important point is that taxonomic annotation of bacteria after DGGE should be updated, as, for instance Lactobacillus plantarum does not exist anymore since the 2020 reclassification, in favor of Lactiplantibacillus plantarum, and so many other former-Lactobacilly A taxonomic note on the genus Lactobacillus: Description of 23 novel genera, emended description of the genus Lactobacillus Beijerinck 1901, and union of Lactobacillaceae and Leuconostocaceae | Microbiology Society (microbiologyresearch.org)
https://www.microbiologyresearch.org/content/journal/ijsem/10.1099/ijsem.0.004107
A4. Thanks. We changed as the journal indicated.
Q5. Line 88: please add information about glutinous and non-glutinous rice (variety, commercial name) and clarify the meaning (vs gluten presence)
A5. Glutinous rice (Oryza sativa var. glutinosa; also called sticky rice is a type of rice grown mainly in Asia including south korea, It has very low amylose content, and is especially sticky when cooked. It is widely consumed across Asia. The reason why it called glutinous in the sense of being glue-like or sticky, and not in the sense of containing gluten (which it does not). We added variety in the text.
Q6. Lines 125-130: incubation conditions do not include microaerophily or anaerobiosis, therefore LAB isolation is partial. Authors should discuss this point. Please also provide antibiotics concentrations.
A6. We did not analyzed anaerobes due to Gayangju fermentation usually done open space with Korean traditional jar. We believe this method enough to understand Gayangju fermentation. We inserted antibiotics concentration.
Q7.Figure 4: please discuss total bacterial counts being lower than LAB-only counts.
A7. There is actually no differce significantly.
Q8. Line 251: please add a reference
A7. Thanks. We added.
Q9 Lines 276-277: please add proper references.
A9 Thanks. References are exists at L282.
Q10 Lines 294-300: it is not clear how this part of discussion brings value to the paper. Since the literature about non-Saccharomyces yeasts in wine is huge and much wider than the single 11-years-old reference cited, either the Authors provide a recent and insightful reading of the point and clearly link it with their results, or (suggested) remove this part.
A10. We removed the sentence L294-300. Thanks.
Q11 Line 334: no statistics appear in the paper. Please explain how significance is determined.
A11 As shown in Materials and methods, Statistical evaluation was performed using the one-way ANOVA, and significant difference was determined using Duncan’s multiple range tests at p < 0.05. The correlation between variables was determined by Pearson’s correlation analysis.
Reviewer 3 Report
The paper reports the study about the Microbial Diversity and Volatile Flavor Changes during Gayangju Fermentation.
This study can be of good interest for the reader of the journal and the methodological approach appears as consistent, as well as the presented processes.
In my opinion, the paper can be some changes, according to the following indications:
Ø In the materials and methods, the procedures must be more descripted, it is important to reported if during the fermentation process there is a mixing end how the temperature is controlled.
I suggest to insert a clear description of process in terms of plant and control systems.
Ø In the results, it is necessary to present they in a schematic form to create a simple description and a clearly of results.
I think that they should be re-written in more clear description.
Ø The conclusions are very synthetic too.
I think that they should be re-written by indicating the main conclusions achieved in this study in very clear form.
Author Response
To Reviewer 3
Q1. The paper reports the study about the Microbial Diversity and Volatile Flavor Changes during Gayangju Fermentation.This study can be of good interest for the reader of the journal and the methodological approach appears as consistent, as well as the presented processes.In my opinion, the paper can be some changes, according to the following indications:
A1. Thanks. we tried to improve our manuscript as best as we can do.
Q2. In the materials and methods, the procedures must be more descripted, it is important to reported if during the fermentation process there is a mixing end how the temperature is controlled. I suggest to insert a clear description of process in terms of plant and control systems.
A2. Gayangju fermentation usually done at 20-25 degree as shown in Figure 1.
Q3. In the results, it is necessary to present they in a schematic form to create a simple description and a clearly of results.I think that they should be re-written in more clear description.
A3 We revised our manuscript in detail. Thanks.
Q4 The conclusions are very synthetic too.I think that they should be re-written by indicating the main conclusions achieved in this study in very clear form.
A4. Thanks. We rewrote conclusion as your comments.
Round 2
Reviewer 1 Report
Several typos need to be corrected, e.g. line 31, line 361.
As figure 5. demonstrated, the DGGE analysis seemed to be carried out one sample per each time point, and no parallel group was conducted. please suggest if any triplicates were made.
In discussion part, it would be interesting if author could compare the change in microbial, physiochemical index and aroma compounds together and try to find the connection between them, as suggested before.
Author Response
To Reviewer 1
Q1. Several typos need to be corrected, e.g. line 31, line 361.
A1. We are very sorry. We corrected them. Thanks.
Q2. As figure 5. demonstrated, the DGGE analysis seemed to be carried out one sample per each time point, and no parallel group was conducted. please suggest if any triplicates were made.
A2. We did the DGGE analysis with three samples per each same time point of different fermentation sample and we select most clear image as most of researchers did. Concerning on DGGE analysis, no researches did parallel group analysis within our knowledge.
Q3. In discussion part, it would be interesting if author could compare the change in microbial, physiochemical index and aroma compounds together and try to find the connection between them, as suggested before.
A3. Exactly. However, it was very difficult to conclude effect of microbial change affecting physiochemical index and aroma compounds together since we did not try to evaluate each microbial properties during Gayangju fermentation. Thus, we inserted this previously: “Further studies regarding the relationships between the flavor profiles and the specific microbes can bring us closer to improving the quality of Gayangju and the efficiency of its production.”
Reviewer 2 Report
Dear Authors,
thank you for your comments, that substantially improved the paper or its comprehension, in particular about the 3 jar fermentors.
Here one last, but still very important, remark:
Authors wrote:
A11 As shown in Materials and methods, Statistical evaluation was performed using the one-way ANOVA, and significant difference was determined using Duncan’s multiple range tests at p < 0.05. The correlation between variables was determined by Pearson’s correlation analysis.
R: this is fine. Therefore, ANOVA outputs should be displayed (letters, asterisks, other) in the tabled or graphed data, and then discussed. Significance of differences is never clearly displayed within data (there is only a "c" letter at the apex in a Table, seems a leftover...)
Author Response
To. Reviewer 2
Q1. As shown in Materials and methods, Statistical evaluation was performed using the one-way ANOVA, and significant difference was determined using Duncan’s multiple range tests at p < 0.05. The correlation between variables was determined by Pearson’s correlation analysis.
R: this is fine. Therefore, ANOVA outputs should be displayed (letters, asterisks, other) in the tabled or graphed data, and then discussed. Significance of differences is never clearly displayed within data (there is only a "c" letter at the apex in a Table, seems a leftover...)
A1. Thanks for your response on statistical evaluation. As your indication we corrected the Table. Thanks.